# Prospects of Using Pseudobrookite as an Iron-Bearing Mineral for the Alkaline Electrolytic Production of Iron

**DOI:** 10.3390/ma15041440

**Published:** 2022-02-15

**Authors:** Daniela V. Lopes, Aleksey D. Lisenkov, Luís C. M. Ruivo, Aleksey A. Yaremchenko, Jorge R. Frade, Andrei V. Kovalevsky

**Affiliations:** 1Department of Materials and Ceramic Engineering, CICECO—Aveiro Institute of Materials, University of Aveiro, 3810-193 Aveiro, Portugal; lisenkov@ua.pt (A.D.L.); luis.ruivo@ua.pt (L.C.M.R.); ayaremchenko@ua.pt (A.A.Y.); jfrade@ua.pt (J.R.F.); akavaleuski@ua.pt (A.V.K.); 2Department of Environment and Planning & Centre for Environmental and Marine Studies (CESAM), University of Aveiro, Campus Universitário de Santiago, 3810-193 Aveiro, Portugal

**Keywords:** Fe_2_TiO_5_, rutile, anatase, hematite, cathodic reduction, electrowinning

## Abstract

The alkaline electrolytic production of iron is gaining interest due to the absence of CO_2_ emissions and significantly lower electrical energy consumption when compared with traditional steelmaking. The possibility of using an iron-bearing pseudobrookite mineral, Fe_2_TiO_5_, is explored for the first time as an alternative feedstock for the electrochemical reduction process. To assess relevant impacts of the presence of titanium, similar electroreduction processes were also performed for Fe_2_TiO_5_·Fe_2_O_3_ and Fe_2_O_3_. The electroreduction was attempted using dense and porous ceramic cathodes. Potentiostatic studies at the cathodic potentials of −1.15–−1.30 V vs. an Hg|HgO|NaOH reference electrode and a galvanostatic approach at 1 A/cm^2^ were used together with electroreduction from ceramic suspensions, obtained by grinding the porous ceramics. The complete electroreduction to Fe^0^ was only possible at high cathodic polarizations (−1.30 V), compromising the current efficiencies of the electrochemical process due to the hydrogen evolution reaction impact. Microstructural evolution and phase composition studies are discussed, providing trends on the role of titanium and corresponding electrochemical mechanisms. Although the obtained results suggest that pseudobrookite is not a feasible material to be used alone as feedstock for the electrolytic iron production, it can be considered with other iron oxide materials and/or ores to promote electroreduction.

## 1. Introduction

Industries such as iron and steel production represent a crucial sector in the global market. Around 70% of the steel manufactured worldwide is still produced by the carbothermal reduction route. This traditional route of converting iron oxides into metallic iron is responsible for 7 to 9% of global CO_2_ direct emissions [1,2]. Apart from molten oxide electrolysis (MOE) [3,4], electrochemical reduction in alkaline media arises as an interesting lean-CO_2_ technology for iron production and steelmaking, considering future industrialization [5,6,7,8,9]. The absence of greenhouse gas emissions, the relatively low temperature (~100 °C) used and simultaneous hydrogen and oxygen production due to the water splitting are attractive features of such electrowinning technology. Significant benefits are expected from the integration of this approach with intermittent renewable energy, considering hydrogen production and storage. Moreover, around 13 GJ/ton of energy are consumed as compared to 19 GJ/ton by the traditional steelmaking route [10,11]. The strong alkaline electrolytes are preferred to the acidic ones, where lower current efficiency is observed due to the looping between iron valence states [7,12] and lower overvoltages for hydrogen evolution, along with less restrictive requirements to manufacture materials in the case of alkaline electrolytes.

Hematite is the most studied raw material for electroreduction to Fe^0^. Studies have been performed both in suspensions [5,13,14] and bulk ceramic cathodes [6,7,8,15]. Despite the associated low electrical conductivity, electrodeposition and electroreduction from Fe_2_O_3_ is proved to be feasible, with Faradaic efficiencies higher than 70% in both modes. Studies on Fe_3_O_4_ [9,14] and FeOOH [14] were also performed, showing, however, generally lower efficiencies. Different feedstocks for the electrochemical reduction for steelmaking are now under consideration, including iron-rich waste such as red mud from the aluminium industry [16,17]. The role of some specific elements like Al [18,19] and Mg [20] on the reduction mechanisms of hematite- or magnetite-based compounds were also studied. Recently, the electroreduction of titanium-containing magnetite, Fe_3–x_Ti_x_O_4_, using ironsands as a feedstock, was also studied [21]. However, relatively low titanium content was considered in that work. In fact, all iron feedstocks relevant for industrial processing contain phase impurities and non-conductive components, which produce hardly predictable effects during electroreduction. Relevant iron oxide-based minerals can be also considered as an alternative feedstock, due to their abundance in the Earth’s crust. Natural iron-titanium oxides, such as pseudobrookite (Fe_2_TiO_5_), are an interesting raw material with a substantially higher Ti content (~33 wt.% of TiO_2_). Fe_2_TiO_5_ is the most stable iron-titanate phase with an n-type semiconductor behaviour, showing suitable properties to be used as a catalyst in biomass gasification [22] due to its compositional flexibility and redox changes, and its capacity to absorb in the visible light spectrum allows it to be used as a photocatalyst [23,24]. It is also used as an electrocatalyst by designing from it defects-rich heterostructures [25] and as a gas sensor [26], since it is a metal oxide semiconductor material or ceramic pigments [27] due to its presence in glassy coatings and glazes after firing at high temperatures. Synthesis of phase-pure Fe_2_TiO_5_ can be performed mainly by solid-state reaction [23,28,29], hydrothermal synthesis [28] and sol-gel [23,24], among others, while the design of highly porous Fe_2_TiO_5_ ceramics seems to be lacking in the literature. The present work explores, for the first time, the use of an iron-bearing mineral such as pseudobrookite as a raw material for iron production and aims to study the impact of higher titanium content on the electroreduction mechanism.

## 2. Materials and Methods

The precursor materials included hematite (Gute Chemie, abcr GmbH, Karlsruhe, Germany, 99.8%) and titanium (IV) oxide (rutile phase, Alfa Aesar, Tewksbury, MA, USA, 99.8%) powders. The powders were ball-milled with ethanol (AGA—Álcool e Géneros Alimentares S.A., Lisbon, Portugal, 99.5%) for 4 h to improve the homogeneity and decrease the particle size of the mixture, followed by oven drying at 60 °C for 24 h. Two types of samples were prepared, dense (Fe_2_TiO_5_ composition) and porous (compositions such as Fe_2_TiO_5_, Fe_2_TiO_5_·Fe_2_O_3_ and Fe_2_O_3_). Dense, disk-shaped Fe_2_TiO_5_ samples were compacted by uniaxial pressing and sintered at 1000 °C for 2 h by microwave irradiation, using a PYRO Microwave Muffle Furnace T480 (Milestone, Sorisole, Italy) with 15 °C/min of heating and cooling rates. The porous ceramics of Fe_2_TiO_5_, Fe_2_TiO_5_·Fe_2_O_3_ and Fe_2_O_3_ were prepared by emulsification with the liquid paraffin method (Valente e Ribeiro Lda., Lisbon, Portugal) from the mixture of powders in a water suspension (50 vol.% of solids) or from the Fe_2_O_3_ powder in water (40 vol.% of solids), according to [30]. Porous samples were first calcined at 500 °C for a complete elimination of the organic phase (1 °C/min until 300 °C, 3 h of dwell; 2 °C/min until 500 °C, 1 h of dwell; 5 °C/min of cooling rate), followed by firing at 1100 °C for Fe_2_O_3_ and 1150 °C for the remaining cases, all with 5 °C/min of heating and cooling rates, and 20 min of dwell. Open porosity of all samples was estimated by the Archimedes method, as described in [8,31].

A three-electrode PTFE electrochemical cell was used for all the electrochemical tests at 80 °C, using 10 M NaOH as electrolyte (100 mL). A Hg|HgO|NaOH (1 M) (+0.098 V vs. standard hydrogen electrode, CH Instruments, Inc., Austin, TX, USA) was used as a reference electrode (RE) connected to the electrochemical cell by a Luggin capillary. A spiral platinum wire (7.40 cm^2^) was used as the counter electrode (CE). The experimental conditions used during the electrochemical reduction tests were selected based on previous results [8,9,18]. The tests were performed for Fe_2_TiO_5_, Fe_2_TiO_5_·Fe_2_O_3_ and Fe_2_O_3_ pellets (2–3 mm thickness, ~1.1 cm^2^ of area), which were used as ceramic cathodes. These pellets were glued to a Ni foil with silver paste (Agar Scientific, Essex, UK). To prevent unwanted electrochemical contributions, the Ni plate was painted with lacquer (Lacomit Varnish, Agar Scientific, Essex, UK). The configuration of the bulk working electrode (WE) was similar to the one described as NFAg-R in our previous work [18]. The schematic electrochemical cell used for the reduction tests in pellets is described in Figure 1. For comparison purposes, the Fe_2_TiO_5_ composition was crushed into powder (<17 µm) in order to prepare alkaline ceramic suspensions (100 g/L, 10 M NaOH). The alkaline suspensions were magnetically stirred (100 rpm) and used as an electrolyte in the process of electrodeposition of Fe^0^ on an iron nail (WE, 1 cm^2^). In suspension mode, the bare iron nail was placed in the cell instead of the Ni plate + ceramic cathode and the electrolyte was filled with the iron oxides suspension. An Autolab PGSTAT302N potentiostat (Metrohm, Herisau, Switzerland) was used for cyclic voltammetry studies (−1.2 V to 0 V; 10 mV/s) and for the electrochemical reduction under galvanostatic mode (1 A/cm^2^) or at potentiostatic mode at potentials such as −1.30 V, −1.20 V and −1.15 V. All samples were washed with ethanol after the electrochemical reduction/deposition and dried in a vacuum desiccator to avoid oxidation in air. Faradaic efficiencies were estimated considering the mass difference of each WE before and after the electroreduction, and also attending to the total electric charge passed in the electrochemical cell. Microstructural features of all ceramic cathodes before and after the electrochemical studies were assessed by a scanning electron microscope (SEM), Hitachi SU-70 (Hitachi High-Technologies Corporation, Tokyo, Japan), coupled with energy disperse spectroscopy analysis (EDS), Bruker Quantax 400. Phase composition was studied by X-ray diffraction (XRD) analysis, using a PANalytical XPert PRO diffractometer (Malvern Panalytical, Malvern, UK), CuKα radiation, 2θ = 10°–80°, with a graphite monochromator, with the software Panalytical HighScore Plus 4.7 (PDF-4).

## 3. Results and Discussion

### 3.1. Structural and Microstructural Features of the Ceramic Cathodes

Previous studies have convincingly demonstrated the importance of porosity and a percolating porous network, facilitating electrolyte access and electrochemical reduction [8,18]. SEM micrographs of the porous ceramic cathodes prepared by the emulsification of aqueous suspension of Fe_2_O_3_-TiO_2_ powders with liquid paraffin are shown in Figure 2, in comparison with the dense Fe_2_TiO_5_ cathode. The formation of highly porous cellular ceramics structures with interconnecting channels is clearly visible for the typical calcined green Fe_2_TiO_5_ samples (Figure 2A). However, significant microstructural evolution is observed after firing at 1150 °C (Figure 2B), showing the formation of thicker walls composed of elongated Fe_2_TiO_5_ grains. This type of microstructure is typical for pseudobrookite-based ceramics, where the highly anisotropic structure promotes the formation of grains with a high aspect ratio, accompanied with significant thermal stresses and formation of microcracks due to anisotropic thermal expansion [32,33]. Such microstructural reorganisation leads to partial collapse of the interconnecting channels and a significant decrease in mechanical strength of the obtained ceramic cathodes. Thus, further firing processes were performed at lower temperatures (1000 to 1150 °C). Ceramic cathodes with various composition and microstructures to ensure a reasonable comparison were prepared, including dense Fe_2_TiO_5_ cathodes (Figure 2C); porous Fe_2_O_3_ (Figure 2D) and porous Fe_2_TiO_5_·Fe_2_O_3_ cathodes (Figure 2E). The average diameter of the cell cavities corresponded to ~10 µm. EDS mapping results show a uniform distribution of Fe_2_TiO_5_ (blue) and Fe_2_O_3_ phases in porous composite Fe_2_TiO_5_·Fe_2_O_3_ cathodes (Figure 2F), confirmed by the XRD results (Figure 3). The addition of Fe_2_O_3_ helps to maintain a well-defined cellular structure after firing. This is illustrated by the comparison of the green ceramics (Figure 2A), where the initial porous skeleton is not yet affected by the grain growth, with the fired Fe_2_TiO_5_ showing almost collapsed cellular structure (Figure 2B) and Fe_2_TiO_5_·Fe_2_O_3_ with clearly observable cellular pores and interconnection channels. Open porosity levels of the cellular samples amounted to 72%, 70% and 56% (vol%), respectively, for Fe_2_TiO_5_, Fe_2_TiO_5_·Fe_2_O_3_ and Fe_2_O_3_ compositions, in contrast with the dense Fe_2_TiO_5_ cathodes (<3% vol.). The formation of single-phase Fe_2_TiO_5_ for both porous and dense cathodes, as well as of Fe_2_O_3_ for the hematite porous cathode is confirmed by XRD (Figure 3). The obtained results show that the selected suspension emulsification conditions allow the preparation of pseudobrookite-based cellular ceramics with a designed porosity, adequate for further electroreduction studies.

### 3.2. Electrochemical Reduction of the Fe-Ti-Bearing Ceramic Cathodes

#### 3.2.1. Studies of the Relevant Redox Processes by Cyclic Voltammetry

General trends of the redox mechanisms occurring in the ceramic cathodes during the electrochemical studies can be obtained from the cyclic voltammograms (CV) represented in Figure 4. The CV analyses were recorded before (t = 0 h) and after (t = 5, 7 h) the electrochemical reduction of the ceramic cathodes, using a scan rate of 10 mV/s. Much lower current densities are observed before the reduction for all the studied cases, due to the generally low electrical conductivity of the samples, in contrast with the plots obtained after reduction, already suggesting the presence of more conductive phases. One should also consider the improvement of the electrical conductivity of the Fe_2_O_3_ cathode (10^−14^ S/cm for Fe_2_O_3_ at room temperature [34,35,36]) when combined with titanium, e.g., in the form of Fe_2_TiO_5_ [29,37,38,39]. Dense pseudobrookite samples demonstrate a better initial electrochemical behaviour, showing higher current densities (Figure 4A) as compared to porous samples. This generally suggests that, at the initial stage, the electrochemical processes are likely limited by the bulk electrode conductivity. The cathodic (C) and anodic (A) peaks/shoulders of the reduction and oxidation of the iron species, correspondingly, are listed in Table 1. Generally, the reduction of Fe(III) to Fe(II) species in all the compositions tested takes place around ~−1 V (C_1_) (Figure 4A,C,E), while the reduction to metallic iron occurs in a superimposed region of the voltammogram associated with the hydrogen evolution reaction (HER) due to the water splitting above ~−1.1 V. HER actively competes for the cathodic current during Fe^0^ formation, decreasing considerably the Faradaic efficiencies, as observed in several works [8,9,13,14,19,20], and it is also responsible for the collapse of ceramic cathodes, as in [18]. The present results confirm the non-direct reduction of hematite-based ceramics to metallic iron, involving a reduction of Fe(III) to Fe(II) aqueous species, where Fe_3_O_4_ is usually a well-established intermediate Fe(III)/Fe(II) phase, in accordance with Pourbaix diagrams [40,41]. Moreover, Fe(II) aqueous species such as Fe(OH)_3_^−^ and mainly HFeO_2_^−^ from the reductive dissolution of Fe_3_O_4_ might be present at the used temperature. The presence of Fe(OH)_2_ is debatable at temperatures higher than 65 °C [42], but it is still considered in several studies [9,43,44]. One can also consider the dissolution of some Fe(III) species to Fe(OH)_4_^−^ anions and its reduction to Fe(OH)_3_^−^ in such strong alkaline media and temperature (~100 °C) [45].

A less cathodic C_1_ shift of ~0.05 V is observed when comparing dense and porous Fe_2_TiO_5_ samples before reduction (Figure 4A). As was previously demonstrated, in porous insulating Fe_2_O_3_ cathodes, the electroreduction starts in the vicinity of the current collector and then progresses to the bulk, promoted by the formation of more conductive phases (Fe_3_O_4_/Fe). Thus, the electron transfer is facilitated in the vicinity of the current collector, and this might be responsible for the earlier onset of the Fe(III) to Fe(II) reduction in the case of the porous samples. However, it is possible to distinguish Fe^0^ formation from the HER current contribution only in the porous samples. The Fe^0^ shoulders (C_2_) are registered around cathodic potentials between −1.13 V and −1.15 V, in agreement with previous studies on porous hematite and hematite-based ceramics under similar experimental conditions [8,18]. Thus, the cathodic potential of −1.15 V was chosen for further electrochemical tests, despite the wide cathodic range observed in the C_2_ shoulders. It should be noticed that the studies on the electrochemical reduction of titanomagnetite sands showed higher cathodic peaks at around −1.2 V vs. (Hg|HgO) when using more concentrated alkali electrolytes (18 M) [21]. The decrease in overall Ti content from Fe_2_TiO_5_ to Fe_2_TiO_5_·Fe_2_O_3_ and to Fe_2_O_3_ could be responsible for the slight shift to less cathodic polarization (~0.02 V for C_2_). In fact, this potential shift might be contributed by the different effective Fe^3+^/Fe^2+^ concentrations in the samples. Moreover, a combined effect from the presence of Ti as non-soluble species (discussed later on) might lead to blocking of the active electroreduction sites, delaying the electrochemical reduction. During oxidation, one can observe anodic peaks at around ~−0.88 V (A_1_) and below −0.7 V (A_2_) due to the oxidation of Fe^0^ to Fe(II) species, and to Fe(III) species (Fe_3_O_4_ and FeOOH), respectively.

When considering the voltammograms after the electroreduction of the ceramic cathodes to Fe^0^ (Figure 4B,D,F), broader peaks were observed for the Fe_2_TiO_5_ and Fe_2_TiO_5_·Fe_2_O_3_ cathodes but not for the case of Fe_2_O_3_. In fact, C_1_ (Figure 4F) is less pronounced (−7.98 mA/cm^2^) due to lower content of Fe(II) species in the sample, while C_2_ is much better defined, indicating higher conversion of Fe(II) species to Fe^0^ in the sample. Also, a third anodic peak (A_0_) around −1.04 V was observed. Since A_0_ is positioned immediately after the potential corresponding to Fe^0^ formation and HER only takes place at more negative potentials, one expects it to be ascribed to some minor Fe^0^ oxidation to Fe(II) species, further continued in the process reflected by A_1_ peak, as in [9,20]. The high conversion to Fe^0^ in the Fe_2_O_3_ samples can also justify the significant shift in the C_1_ peak to less cathodic potentials (0.1 V) after the electrochemical reduction. On the other hand, broader peaks were found after the eletroreduction for the Fe-Ti ceramic cathodes (Figure 4B,D), most likely due to the slow kinetics of the conversion of iron species when blocked with Ti low-soluble species, similar to a Mg-containing ferrospinel cathode [20].

#### 3.2.2. Chronoamperometric Studies of the Dense Fe_2_TiO_5_ Ceramic Cathodes

Dense single-phase pseudobrookite cathodes were subjected to chronoamperometric studies aiming at the complete reduction to Fe^0^ (Figure 5). A multi-step experiment was performed by applying a cathodic potential of −1.15 V, followed by an increase to −1.20 V, due to the absence of any noticeable impact in the current density at low polarizations (Figure 5A), for a maximum period of 7 h. 

Relatively low current densities were also reached when a second single experiment was performed, by increasing the cathodic polarization to −1.30 V (<30 mA/cm^2^) for 5 h, also showing noise in the chronoamperometric curve, most likely due to the effects exerted by hydrogen bubbles at the cathode surface. 

Post-mortem studies of the ceramic cathodes clearly show signs of delamination and cracks regardless of the cathodic polarization applied. This effect was also observed when decreasing the polarization down to −1.08 V (results not shown), revealing similar cracks between a “greyish” and “reddish” zone, assumed as an Fe_2_TiO_5_ electroreduction product and the initial pseudobrookite structure, correspondingly. The “greyish” zone was found near the current collector (CC), while the “reddish” was found on the opposite side, as observed on the inset image in Figure 5A. 

SEM mapping images show the different morphology of both areas, separated by a surface line (SL). Both areas show the presence of octahedral crystals enriched with iron, which likely correspond to Fe_3_O_4_, in agreement with the XRD diffractogram (Figure 5B). The growth of these crystals apparently results in cracks and delamination in the SL area, promoted by significant structural and microstructural rearrangements during the transformation phase of highly anisotropic pseudobrookite into magnetite. Near CC, magnetite crystals surrounded by a nanostructured amorphous anatase phase (needles marked as blue in the mapping images) were found. In fact, this coverage might be responsible for blocking the possible reduction to Fe^0^ since the conductive Fe_3_O_4_ crystals become isolated by the insulating phase. Some of the Fe_3_O_4_ crystals show slits and significant porosity in their interior (Figure 5C). Although exact reasons for such behaviour are still unclear, these might be related to excessive hydrogen evolution in the regions where a good electrical connection with the bulk cathode is still maintained, while another part of the cathode surface becomes deactivated due to the formation of insulating TiO_2_ phases.

The electrochemical reduction of the Fe_2_TiO_5_ composition is significantly limited due to the described phenomena. By using porous ceramic cathodes instead of dense ones, one expects to facilitate the electrochemical reduction to Fe^0^ due to the percolation of the electrolyte into the cathode bulk, as already proved in previous works [8,18,20]. Moreover, the cracks’ propagation, similar to that presented in inset in Figure 5, followed by losses of electrical contact, is expected to be limited in porous ceramics, as compared to dense ones.

#### 3.2.3. Chronoamperometric Studies of the Porous Cathodes

The results on cathodic reduction of porous cathodes and corresponding microstructural evolution are represented in Figure 6. The electroreduction was performed in potentiostatic mode during 7 h, with polarization of −1.15 V (Figure 6A) and −1.30 V (Figure 6B). The selection of cathodic potential of −1.15 V was based on the results obtained in Table 1 (peak C_2_). Previously, the electrochemical reduction of porous cathodes was proved to be more effective when compared to dense ones. The presence of pores combined with a suitable interconnectivity between them facilitates the access of the electrolyte and migration of all the ionic species involved in the reduction to Fe^0^, in agreement with previous studies [8,15,18].

Both Fe_2_TiO_5_ and Fe_2_TiO_5_·Fe_2_O_3_ show similar electrochemical performance, with slower reduction progress after 3 h (65 mA/cm^2^) and 1 h (60 mA/cm^2^), respectively. Lower currents observed during the initial stage of the electroreduction of Fe_2_O_3_ are likely a result of a higher resistivity of Fe_2_O_3_ as compared to Fe_2_TiO_5_. During the first 0.3 h, the cathodic current even decreases due to the processes related to the formation of a stable interface between the current collector, cathode pellet and electrolyte, promoted by the electrolyte entrance in the porous cavities. However, Fe_2_O_3_ cathodes show continuous progress in the electroreduction, accompanied by a gradual increase of the electrical current, reaching 127 mA/cm^2^ in 7 h due to the formation of more conductive phases, including Fe^0^. The reduction is still incomplete after 7 h, as indicated by the absence of a current density plateau after the 7 h of the experiment. The plateau presence and current density decrease, which usually occur after the complete reduction of the cathode to Fe^0^ can be explained within the three-phase interline (3PI) model proposed by [15,45]. Titanium-bearing cathodes reach stable values of the current much faster than one could expect assuming a high Faradaic efficiency of the reduction under the used experimental conditions. These results suggest the presence of notable kinetic limitations impeding the bulk electroreduction of Ti-bearing cathodes, in accordance with the above-mentioned CV results.

In order to understand better the electrochemical behaviour of Ti-bearing cathodes and establish possible limiting factors, the chronoamperometric studies were performed at fairly more negative cathodic potentials of −1.30 V (Figure 6B). While the plateau can be observed for all tested compositions, with at least 10 times higher current densities, one cannot observe the expected decrease of the current density, indicating complete electroreduction. The reproducibility of the results is, however, quite poor, as illustrated by the green and blue curves corresponding to two different trials of the electroreduction of Fe_2_TiO_5_ samples under identical conditions. In the Fe_2_TiO_5_·Fe_2_O_3_ case, the cathode pellets were partially disintegrated and cracked, as shown in the inset (Figure 6B), likely due to significant mechanical stresses exerted by transformation of phases and excessive (and probably localised) hydrogen evolution. These factors lead to hardly predictable variations of the cathode surface area available for electroreduction. However, the above troubles were never an issue for Fe_2_O_3_ cathodes, regardless of the applied potentials.

Corresponding SEM micrographs are shown in Figure 6C–F. The reduction of Fe_2_O_3_ occurs as expected, with dendritic Fe^0^ growth (Figure 6C) observed at both potentials used for the cathodic reduction. In the case of Fe_2_TiO_5_·Fe_2_O_3_ cathodes, octahedral crystals (most likely Fe_3_O_4_) appear inside the porous structures (Figure 6D), indicating the incomplete electroreduction. On the other hand, the post-mortem analysis of the Fe_2_TiO_5_ cathodes reveals interesting results, showing the formation of few dendritic Fe^0^ growths near CC, when −1.30 V was applied in Figure 6E (SEM image of the sample from the blue curve in Figure 6B). However, the results are poorly reproducible, since another sample tested under similar conditions showed mostly the formation of octahedral crystals in the pores (Figure 6F, SEM image of the sample from the green curve in Figure 6B). Still, for this sample, one observed the presence of micro-rods enriched with iron, which, to a certain extent, resemble the structure of the typical Fe^0^ dendrites. The latter suggests that complete electroreduction of the Fe_2_TiO_5_ cathodes under the discussed conditions might still be possible but is progressively blocked by the simultaneous formation of TiO_2_ phases, limiting the electrical contact between Fe-enriched conducting products.

XRD patterns after the electroreduction are shown in Figure 7, in fair agreement with the results presented in Figure 6. The presence of Fe^0^ is obvious in the Fe_2_O_3_ samples (Figure 6A), where around 19 wt% and 81 wt% of Fe^0^ are present when the cathodic potentials of −1.15 and −1.30 V (not shown) were applied, respectively. Magnetite was also found as a secondary phase, but mostly unreduced Fe_2_O_3_ was found, confirming a surface reduction of the pellet, not reaching the core, as mentioned during the chronoamperometry results. Current efficiencies were lower, 20% and 6% for −1.15 and −1.30 V, at 80 °C, when compared with the literature data for porous Fe_2_O_3_ samples [8,15,18], due to the higher cathodic polarization used in this work. When increasing the cathodic polarization, the hydrogen evolution reaction lowers the Faradaic efficiency, despite relatively higher current densities being attained. Concerning the reduction of the Fe_2_TiO_5_ cathodes, metallic iron was only observed in this composition when high cathodic polarizations of −1.30 V were applied (Figure 6B). Fe_2_TiO_5_·Fe_2_O_3_ cathodes (Figure 7C) revealed the absence of Fe^0^. However, both rutile and anatase are found in this sample, likely blocking the electroreduction, as suggested by the microstructural studies. Although this fact actually suggests that the complete electroreduction of Ti-bearing iron oxides might be not feasible under discussed conditions, it may open new prospects for electrochemical conversion of the Ti-bearing minerals into materials with potentially high photocatalytic activity.

In general, the electroreduction of Fe_2_TiO_5_ to Fe^0^ in bulk cathodes appears to be feasible only at high cathodic potentials above −1.30 V. The galvanostatic approach was also attempted and discussed in the next section to bring more insights on the prospects for electrochemical reduction of pseudobrookite.

#### 3.2.4. Galvanostatic Electroreduction

Fe_2_TiO_5_ ceramic cathodes were tested in the same experimental conditions but under a galvanostatic mode using 1 A/cm^2^ current density. 

Figure 8 shows the potential vs. time curve, together with the notable physical changes of the sample, and its XRD pattern. A non-uniform greyish region was found near CC after the electroreduction, corresponding to the end of the experiment reaching the potentials around −1.30 V. A dense Fe^0^ layer of 220 µm was found by Bjäreborn et al. [21] at the surface of titanomagnetite when 1 A/cm^2^ was applied in galvanostatic mode. When higher current densities up to 2.3 mA/cm^2^ were applied, a thicker layer of porous dendritic Fe^0^ was observed (340 µm). 

In the present work, while the reddish part of the sample can be related to the initial Fe_2_TiO_5_ structure (red is the initial colour of pseudobrookite samples), one can detect the presence of Fe^0^ in the greyish section (<1 wt.%), along with Ti-phases (~4 wt.%), as confirmed by the XRD results presented in Figure 8B. Very low Faradaic efficiencies were obtained in these conditions, since HER also played an important role at such high current densities by competing with the cathodic current for the reduction to Fe^0^.

Negligible titanium-phases were observed on the Fe^0^ phase (~1wt.%) in [21], but they were mainly present on the unreduced part of the cathodes. The fate of Ti was assumed to be exsolved as solid inclusions, or to the electrolyte or as phase-segregated solids in the unreduced section of the pellet. Despite the mentioned work involving more concentrated electrolyte and higher temperature for the electrochemical reduction when applying the current density of 1 A/cm^2^, the significant difference with the present work is associated with using much less titania addition. In the present work, the higher titanium content has a stronger impact on the electroreduction process due to rather pronounced segregation of Ti-containing phases, eventually covering the porous structure and blocking further electrochemical reduction.

More insights on the electrochemical reduction mechanism of Fe_2_TiO_5_ can be obtained by analysing the SEM microstructures of the cathodes after the galvanostatic reduction, represented in Figure 9. The SEM images suggest the dissolution–redeposition electrochemical process occurring in the ceramic cathodes. The C_1_ image represents the typical initial pseudobrookite particle with some “needle” nanostructures formed at the surface related to anatase and rutile, as previously shown in Figure 5B. The crystals marked as red represent the formation of intermediate electroreduction product, Fe_3_O_4_, eventually growing and evolving to cluster crystals (C_2_). The next stage of the Fe_3_O_4_ formation is represented by binding those clusters to form larger octahedral crystals that show some “defects” until the final shape is totally formed (C_3_, C_4_). Similar micro-rods marked at red as in Figure 6 also can be observed near nanostructured rutile and anatase (C_5_), possibly indicating the presence of Fe^0^, as confirmed by the XRD diffractogram. The C_4_ image clearly shows the blocking of conducting islands composed of red iron-enriched crystals by the blue insulating TiO_2_ phases. This blocking apparently involves losses of the electrical contact, as discussed before, and hampers the redeposition of iron from dissolved Fe^2+^ species. The HER effect is expected to be more pronounced at higher potentials, causing partial physical removal of the nanostructured TiO_2_ phases by hydrogen bubbles. This explains the Fe^0^ presence when applying higher current densities and potentials (−1.30 V) when compared to −1.15 V, in accordance with the previous XRD diffractograms.

### 3.3. Electrochemical Reduction of Alkaline Fe_2_TiO_5_ Suspensions

The possibility of electroreduction from alkaline Fe_2_TiO_5_ suspensions was also evaluated both in potentiostatic and galvanostatic modes, aiming at strategies for feasible industrial application. In this respect, while the electroreduction of bulk cathodes is expected to provide more trends regarding the process mechanism, the electrolysis of ceramic suspensions appears to be suitable for large-scale iron production. The obtained results are represented in Figure 10A. While applying a cathodic polarization of −1.30 V, notably high currents up to 440 mA/cm^2^ can be reached due to the deposition of a conductive phase, such as Fe^0^, and corresponding increase of the cathode surface area, followed by a small current density decrease. When applying lower potentials (−1.15 V), the current density decreased by around 20% with time, reaching 39 mA/cm^2^. In the galvanostatic mode (Figure 10B), the applied potential decreases to −1.40 V and −1.45 V, for reasons similar to those promoting a current increase in Figure 10A. SEM images of all the obtained deposits (Figure 10C–E) show quite distinct microstructures when compared to [13,19,20]. The usual dendritic shape obtained when using Fe_2_O_3_-based suspensions in similar electroreduction conditions was not observed in any of the cases. Round and dense nodules can be seen instead, apparently formed from several nanometric crystals. This shape is similar to that observed by Feynerol et al. [14] for goethite alkaline suspensions. However, the XRD pattern of the deposits (Figure 10F) clearly prove the presence of Fe^0^ along with traces of Fe_3_O_4_. Ti-containing phases were not found trapped in the deposits.

Moreover, the observed Fe_3_O_4_ phase should not be considered as the common intermediate phase during the reduction in pellets/bulk, but as a phase formed due to the minor oxidation of the samples during their handling. The observed promising results can be explained not only by the use of the high cathodic polarization, when compared with the lower potential-Fe_3_O_4_ region in Pourbaix diagrams at 100 °C [42], but also by the specificity of the electroreduction in suspensions. The mechanism is trigged by some adsorption of the particles on the surface of the WE, combined with the Fe(III) dissolution as hydrated Fe(OH)_4_^−^ anions and its reduction to Fe(II) species as Fe(OH)_3_^−^. Further reduction results in the deposition of Fe^0^ on the WE [14,46,47,48]. When increasing the polarization up to −1.30 V, a much denser Fe^0^ layer was obtained, showing a considerable decrease in the crystal size from 30 µm dispersed nodules to a few clusters (7.5 µm) deposited on a very compact layer of Fe^0^ crystals. The higher polarization during the galvanostatic electrodeposition lead to even smaller < 5 µm sized dispersed nodules. HER has a stronger effect during higher cathodic polarization, and is possibly responsible for removing some of the Fe^0^ deposits. The reason possibly lies in the current density decrease and potential increase closer to the end of the experiments, as shown in Figure 10A,B. In general, the observed morphology differences and considerably lower Faradaic efficiencies when compared with pure Fe_2_O_3_ suspensions might be linked to combined effects of HER and the presence of sodium iron titanate species in the electrolyte [21,49,50]. Around 25%, 5% and 4% of current efficiencies were obtained with experimental conditions of −1.15 V, −1.30 V and 1 A/cm^2^, respectively.

## 4. Conclusions

The electrochemical reduction of a titanium-bearing pseudobrookite mineral, Fe_2_TiO_5_, was attempted both in pellet/bulk ceramic cathodes with high porosity and in suspension modes, under strong alkaline conditions (10 M, NaOH). Particular attention was given to the design of highly porous cathodes by the emulsification with liquid paraffin prior to the electroreduction, promoting contact between the electrode and electrolyte. The impact of the high content of titanium on the electroreduction of iron oxides was explored for the first time, by comparative electrochemical studies of Fe_2_TiO_5_, Fe_2_TiO_5_·Fe_2_O_3_ and Fe_2_O_3_. Partial reduction followed by the formation of the Fe_3_O_4_ crystals in bulk ceramic cathodes is clearly demonstrated by SEM-EDS and XRD studies, a crucial step in the mechanism of reduction. However, further electroreduction was shown to be blocked due to the formation of insulating TiO_2_ phases, as demonstrated by microstructural studies.

The present results demonstrate the possibility of producing metallic iron from Fe_2_TiO_5_ when cathodic polarization higher than −1.30 V are applied or when relatively high cathodic current densities (1 A/cm^2^) are used in galvanostatic mode. Relatively low Faradaic efficiency, hardly reaching 25%, was attained by the electroreduction from suspensions. In general, Faradaic efficiencies of iron reduction are hampered when using high cathodic polarizations due to the competing hydrogen evolution reaction, destroying the Fe_2_TiO_5_·Fe_2_O_3_ cathodes. Despite appearing to be not feasible for industrial steel production by the electrowinning process, natural pseudobrookite and other iron-bearing minerals with lower Ti content still can be considered as a feedstock when mixed with other iron rich-materials for a better efficiency of the process. While affecting negatively the electroreduction process, the promoted formation of nanostructured rutile and anatase may open new strategies for the synthesis of composite materials for photocatalytic and catalytic applications, using abundant minerals feedstock.

## Figures and Tables

**Figure 1 materials-15-01440-f001:**
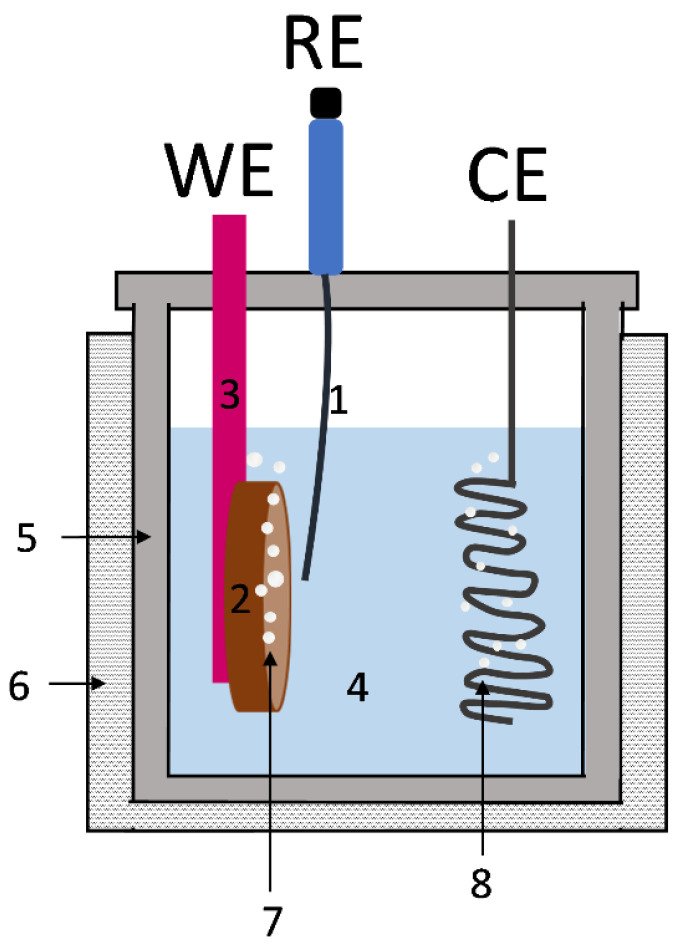
Scheme of the electrochemical cell used for the electroreduction of the samples: WE—working electrode; RE—reference electrode; CE—counter electrode (Pt wire); 1—Luggin capillary; 2—ceramic cathode; 3—Ni plate covered with lacquer (or iron nail in the case of suspensions); 4—electrolyte (NaOH, 10 M); 5—Teflon cell; 6—heating system; 7—hydrogen bubbles; 8—oxygen bubbles.

**Figure 2 materials-15-01440-f002:**
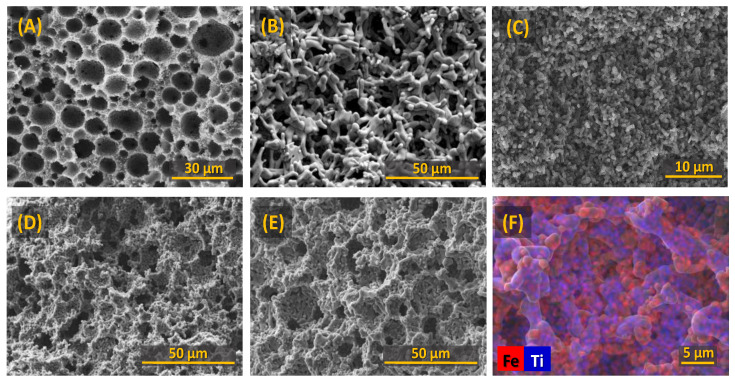
Micrographs of the ceramic cathodes: (**A**) typical cellular Fe_2_TiO_5_ green monolith (500 °C); (**B**) cellular Fe_2_TiO_5_ (1150 °C); (**C**) dense Fe_2_TiO_5_ (1000 °C); (**D**) cellular Fe_2_O_3_ (1100 °C); (**E**) cellular Fe_2_TiO_5_·Fe_2_O_3_ (1150 °C); (**F**) EDS image of cellular Fe_2_TiO_5_·Fe_2_O_3_ (1150 °C).

**Figure 3 materials-15-01440-f003:**
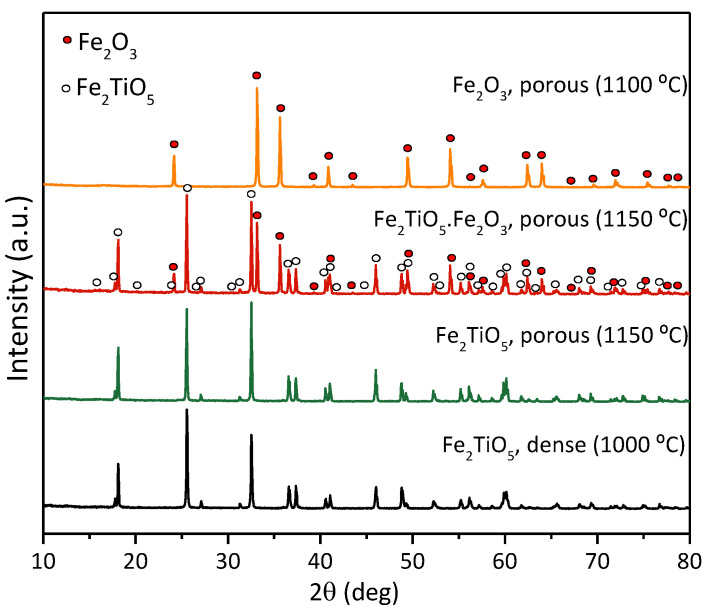
XRD patterns of all the ceramic cathodes and corresponding firing/sintering temperatures.

**Figure 4 materials-15-01440-f004:**
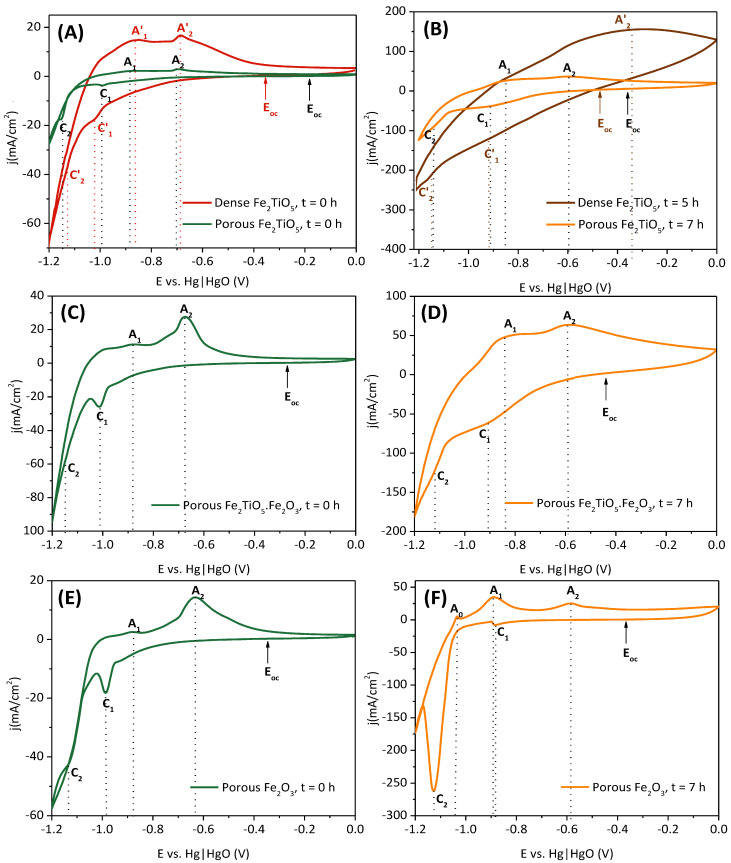
CV plots before (t = 0 h, green or red for dense pellets) and after (t = 5, 7 h, orange or brown for dense pellets) the electrochemical reduction of (**A**,**B**) Fe_2_TiO_5_; (**C**,**D**) Fe_2_TiO_5_·Fe_2_O_3_; (**E**,**F**) Fe_2_O_3_; recorded with 10 mV/s (10 M NaOH, 80 °C).

**Figure 5 materials-15-01440-f005:**
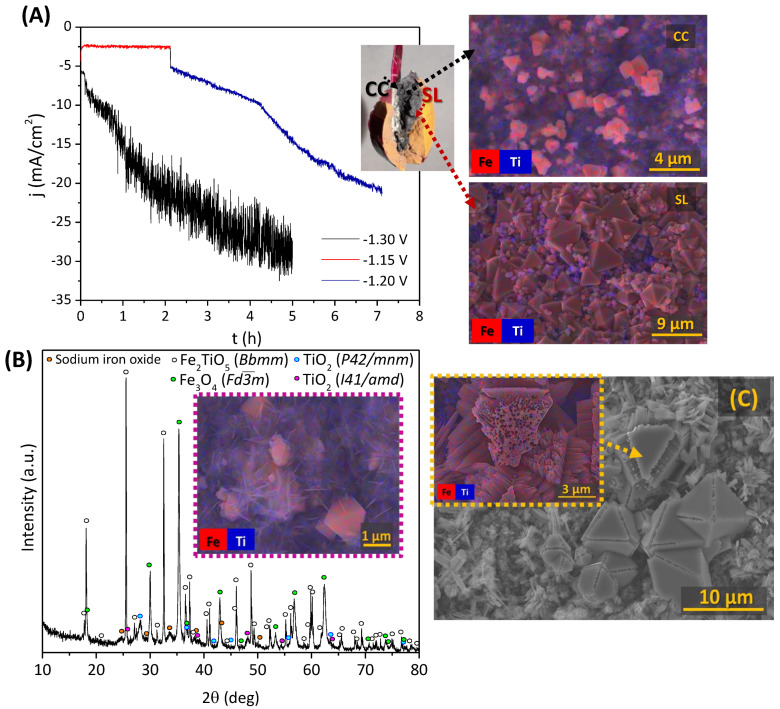
Electrochemical reduction of Fe_2_TiO_5_ dense ceramic cathodes: (**A**) chronoamperometry at −1.30 V, −1.20 V and −1.15 V (10 M NaOH, 80 °C) and EDS images of the Fe_3_O_4_ crystals; (**B**) typical XRD diffractogram after the electroreduction; (**C**) SEM + EDS image of Fe_3_O_4_ crystals formation.

**Figure 6 materials-15-01440-f006:**
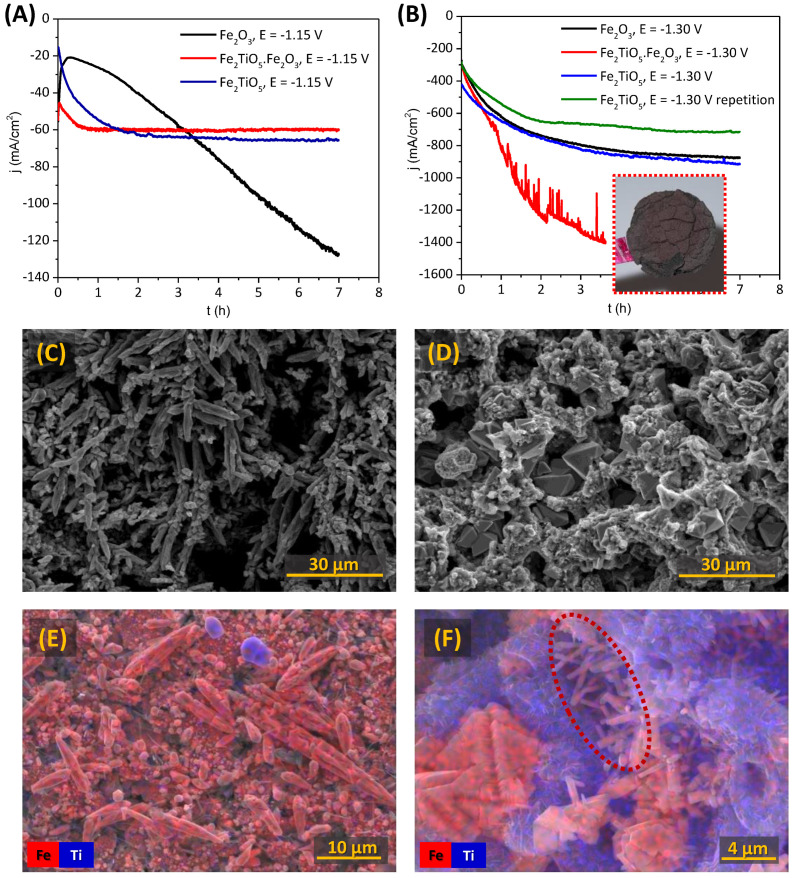
Electrochemical reduction of the porous Fe_2_O_3_, Fe_2_TiO_5_·Fe_2_O_3_ and Fe_2_TiO_5_ ceramic cathodes: (**A**,**B**) chronoamperometry at −1.15 V and −1.30 V, respectively (10 M NaOH, 80 °C); typical SEM and EDS mapping images after the electroreduction of the cathodes of (**C**) Fe_2_O_3_ (potentiostatic mode on −1.15 V); (**D**) Fe_2_TiO_5_·Fe_2_O_3_ (−1.15 V) and (**E**,**F**) Fe_2_TiO_5_ (−1.30 V).

**Figure 7 materials-15-01440-f007:**
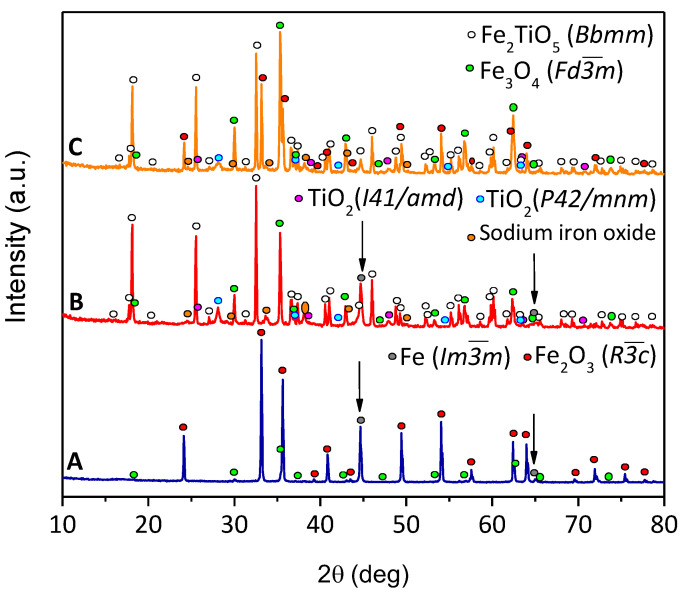
XRD diffractograms of the cathodes after the electrochemical reduction of: (**A**) Fe_2_O_3_ (−1.15 V); (**B**) Fe_2_TiO_5_ (−1.30 V); (**C**) Fe_2_TiO_5_·Fe_2_O_3_ (−1.15 V).

**Figure 8 materials-15-01440-f008:**
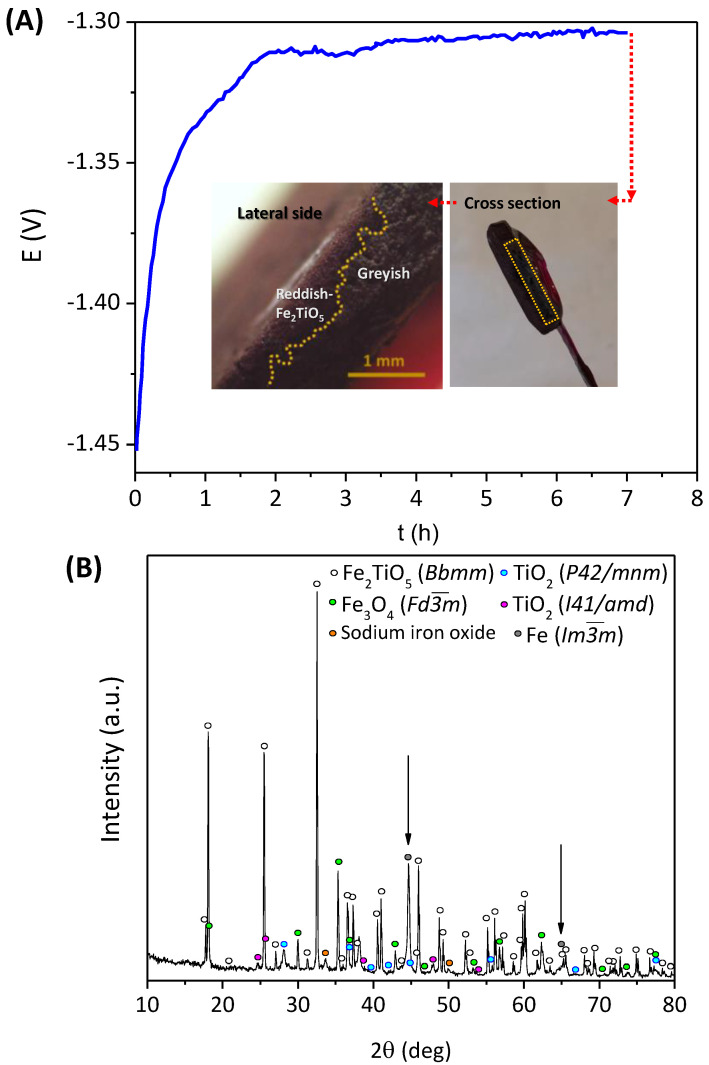
Galvonostatic electroreduction (1 A/cm^2^, 7 h): (**A**) chronopotentiometry results during 7 h, 1 A/cm^2^; (**B**) XRD diffractogram of the final bulk pellet.

**Figure 9 materials-15-01440-f009:**
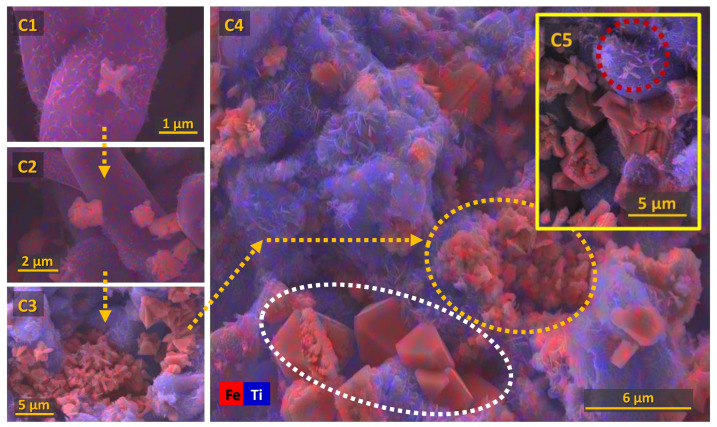
The microstructural evolution from (C1) to (C5) of the ceramic Fe_2_TiO_5_ cathodes reduced under galvanostatic mode, 1A/cm^2^ during 7 h.

**Figure 10 materials-15-01440-f010:**
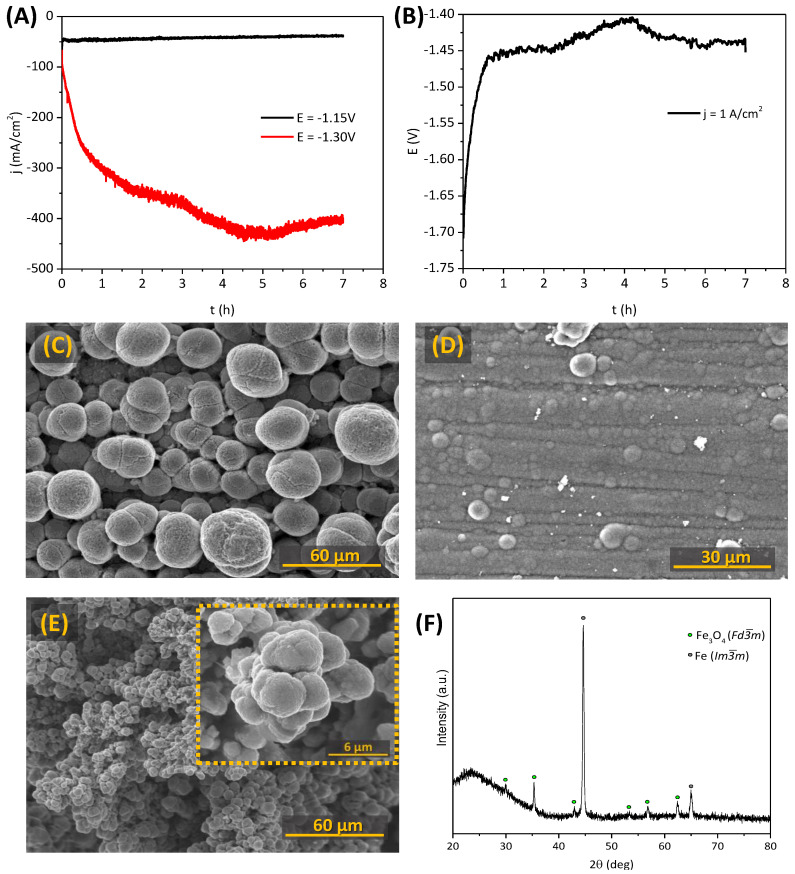
The results obtained for electrochemical deposition from alkaline Fe_2_TiO_5_ suspensions during 7 h (10 M NaOH, 80 °C): (**A**) potentiostatic electrodeposition at −1.15 V and −1.30 V; (**B**) galvanostatic electrodeposition at 1 A/cm^2^; SEM images of the deposits after: (**C**) −1.15 V (potentiostatic); (**D**) −1.30 V (potentiostatic); (**E**) 1 A/cm^2^ (galvanostatic); (**F**) typical XRD pattern of the deposits.

**Table 1 materials-15-01440-t001:** Cathodic and anodic peaks/shoulders from the cyclic voltammograms before and after the electrochemical reduction.

Composition	Before Reduction		After Reduction	
	Cathodic Peaks/Shoulders (V)	Anodic Peaks/Shoulders (V)	Cathodic Peaks/Shoulders (V)	Anodic Peaks/Shoulders (V)
Dense Fe_2_TiO_5_	C′_1_ = −1.04 C′_2_ = −1.14	A′_1_ = −0.87 A′_2_ = −0.69	C′_1_ = −0.92 C’_2_ = −1.15	- A′_2_ = −0.35
Porous Fe_2_TiO_5_	C_1_ = −0.99 C_2_ = −1.15	A_1_ = −0.88 A_2_ = −0.68	C_1_ = −0.91 C_2_ = −1.14	A_1_ = −0.85 A_2_ = −0.59
Porous Fe_2_TiO_5_·Fe_2_O_3_	C_1_ = −1.01 C_2_ = −1.15	A_1_ = −0.88 A_2_ = −0.67	C_1_ = −0.91 C_2_ = −1.13	A_1_ = −0.84 A_2_ = −0.59
Porous Fe_2_O_3_	- C_1_ = −0.98 C_2_ = −1.13	- A_1_ = −0.87 A_2_ = −0.63	- C_1_ = −0.88 C_2_ = −1.12	A_0_ = −1.04 A_1_ = −0.89 A_2_ = −0.59

## Data Availability

Not applicable.

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
