# Peer review of "Prospects of Using Pseudobrookite as an Iron-Bearing Mineral for the Alkaline Electrolytic Production of Iron"

_materials, 2022, doi:10.3390/ma15041440_

Round 1

Reviewer 1 Report

This work describes an interesting study on the alkaline electrolytic production of iron from titanium-bearing pseudobrookite mineral, Fe2TiO5. Presented results are obviously of very early nature and thus deprived of more elaborate and deeper understanding of the possible chemical and electrochemical processes taking part in such a complex system. The authors have worked on the design of highly porous cathodes to be used for the electroreduction of iron (titanium) from oxide suspensions in highly alkaline solutions. The metallic iron was produced from Fe2TiO5 at cathodic potentials higher than -1.30 V or at relatively high cathodic current density (1A/cm2). In the manuscript, the impact of the titanium content on the iron electroreduction was also addressed. However, some of the results and interpretations that this paper provides should be clarified.

  • In general, company names and locations should be provided together with instruments and reagents;
  • Please, alongside the thickness, specify the cathode area employed for the electrochemical tests, this can be mentioned in the Materials and Methods section;
  • Fig.3. and Table 1. The cathodic peak C2 is hardly visible on some CV’s, especially on CV’s on Figs.3.A, C, D... If all tests are performed in controlled conditions, than the chosen potential of cathodic peak C2 should be explained, especially because this cathodic potential was selected for further experimental test.
  • Chronoamperometric studies of the dense and porous Fe2TiO5 ceramic cathodes. Some of the interpretations of the results that this paper provides seem insufficient to support the claims. The exact role of solid-state processes and reduction reactions should be better defined.
  • Another point rises when we compare the results obtained by electroreduction of Fe2TiO5∙Fe2O3 porous cathodes performed in potentiostatic mode. In this study the results regarding electrochemical reduction of the porous Fe2TiO5∙Fe2O3 cathode at -1.30 V are missing. Something should be mentioned about this.

Overall, it’s a good contribution to this field and can be accepted with minor revisions.

Author Response

The authors would like to thank the reviewer for the valuable comments, which helped us to improve the work. The changes made in the text of the manuscript are marked by track changes.

1. In general, company names and locations should be provided together with instruments and reagents;

The company names and their respective locations are now provided through the text in the Materials and Methods section.

2. Please, alongside the thickness, specify the cathode area employed for the electrochemical tests, this can be mentioned in the Materials and Methods section;

The cathodes area was provided in the text, as requested.

3. Fig. 3. and Table 1. The cathodic peak C2 is hardly visible on some CV’s, especially on CV’s on Figs.3.A, C, D... If all tests are performed in controlled conditions, than the chosen potential of cathodic peak C2 should be explained, especially because this cathodic potential was selected for further experimental test.

The authors understand the reviewer’s concern. In fact, C2 was considered a “shoulder” and not a peak. This happened due to a combination of factors such as the superimposition of hydrogen evolution reaction (HER) with the Fe0 formation, with kinetic limitations of the ceramic cathodes in the presence of non-conductive phases (Ti), which widens the cathodic peak. This effect was also observed in several other works in the literature as mentioned in the same section of the manuscript. Nevertheless,  the “shoulder” region is clearly present in all CV curves in the same region (~ -1.13 V to ~ -1.15 V). The authors believe that the variation of ~0.02 V in the discussed range is not critical for Fe0 formation, while only causing the difference in the current densities attained. In fact, the chronoamperometry studies for dense pellets were performed at different potentials, -1.08 V (mentioned in the text but no results were shown due to the low current densities attained), -1.15 V, -1.20 V and -1.30 V. All these studies showed the same outcome: no Fe0 phase was observed in any of the cathodes and the higher polarization lead to a higher noise due to HER. All these potentials cover the range of HER superimposed with Fe0 phase formation.  In porous samples, two cathodic polarizations were chosen, -1.15 V and -1.30 V, for the sake of comparison with the dense samples. Once again, a small variation in the polarization would not be critical for the Fe0 production, since Fe0 was only formed in the Fe2TiO5 cathodes at -1.30 V . We understand that the readers may have a similar concern. Thus, the word “shoulders” instead of peaks and the range between -1.13 V to -1.15 V for the cathodic shoulders was reinforced in the manuscript.

4. Chronoamperometric studies of the dense and porous Fe2TiO5 ceramic cathodes. Some of the interpretations of the results that this paper provides seem insufficient to support the claims. The exact role of solid-state processes and reduction reactions should be better defined.

We do agree that the studies we performed do not give comprehensive answers regarding the exact electroreduction mechanisms. In particular, the potentiostatic measurements were not enough conclusive to ascertain the role of Ti during the reduction, since several issues such as cracks and mechanical disintegration of the cathodes happened through the process. Still, in 3.2.4. section, after the galvanostatic reduction, the  possible mechanisms of the electrochemical reduction process of Fe2TiO5 is discussed by combining SEM/EDS results with XRD diffractograms, against the CV curves obtained, supported by Pourbaix diagrams. Thus, the reduction mechanism can be explained by the following sentece: “The SEM images suggest the dissolution-redeposition electrochemical process occurring in the ceramic cathodes. C1 image represents the typical initial pseudobrookite particle with some “needle” nanostructures formed at the surface related to anatase and rutile, as previously shown in Figure 5B. The crystals marked as red represent the formation of intermediate electroreduction product, Fe3O4, eventually growing and evolving to cluster crystals (C2). The next stage of the Fe3O4 formation is represented by binding those clusters to form larger octahedral crystals that show some “defects” until the final shape is totally formed (C3, C4). Similar micro-rods marked at red as in Figure 6 also can be observed near nanostructured rutile and anatase (C5), possibly indicating the presence of Fe0 as confirmed by the XRD diffractogram. The image C4 clearly shows the blocking of conducting islands composed of red iron-enriched crystals by the blue insulating TiO2 phases. This blocking apparently involves losses of the electrical contact, as discussed before, and hampering the redeposition of iron from dissolved Fe2+ species. HER effect is expected to be more pronounced at higher potentials, causing partial physical removal of the nanostructured TiO2 phases by hydrogen bubbles. This explains the Fe0 presence when applying higher current densities and potentials (-1.30 V) when compared to -1.15 V, in accordance with the previous XRD diffractograms.”.

5. Another point rises when we compare the results obtained by electroreduction of Fe2TiO5∙Fe2O3 porous cathodes performed in potentiostatic mode. In this study the results regarding electrochemical reduction of the porous Fe2TiO5∙Fe2O3 cathode at -1.30 V are missing. Something should be mentioned about this. Fe2TiO5∙Fe2O3.

The electrochemical reduction of the porous Fe2TiO5∙Fe2O3 cathode at -1.30 V is present as the red curve in the Figure 6B. At such high cathodic polarizations (-1.30 V), the mechanical desintegration of the pellets by HER is visible in all replicates trials, as it can be also confirmed by the current density oscillations on the red curve represented and the cracks at the pellet in the inserted figure. Once the pellet is cracked during the electrochemical test, one cannot rely on the results due to the higher exposition of the reduced area to the electrolyte, since the part having the initial composition of the pellet is detached to the electrolyte. The following sentence is given in the manuscript regarding this specific situation: “In the Fe2TiO5·Fe2O3 case, the cathode pellets were partially disintegrated and cracked, as shown in inset (Figure 6B), likely due to significant mechanical stresses exerted by transformation of phases and excessive (and probably localised) hydrogen evolution. These factors lead to hardly predictable variations of the cathode surface area available for electroreduction. However, the above troubles were never an issue for Fe2O3 cathodes, regardless the applied potentials.”.

Reviewer 2 Report

This is an overall well presented,organised  and innovative work. Only a few minor comments from my side to the authors: 

-Page 2, when referencing own work (Ref 18) please state it more clearly in the text

-Page 3 "the electric charge passed in the electrochemical cell" is not presented in the paper

- Page 4 (and elsewhere) Use of the word "guidelines" is confusing.Consider  using 'trend' or similar

-Page 7, section 3.2.2 it is not clear wheather these are the results of a single multi-step experiment or a series of independent experiments. Also the total duration should be mentioned - similar to paragraph 3.2.3

-Page 7: Although exact reasons for such behaviour are still unclear, those might be related to excessive hydrogen evolution in the regions where a good electrical connection with the bulk cathode is still maintained, while another part of the cathode surface become desactivated due to the formation of insulating TiO2 phases. Consider changing those to these and deactivated instead od desactivated

-Page 7 :Thus, the electrochemical reduction of Fe2TiO5 composition is significantly limited due to the mentioned reasons. Consider changing 'mentioned reasons' to 'described phenomena' 

-Page 9:  when compared with the literature data for porous Fe2O3 samples, please provide reference for literature data

-Page 9 :  Metallic iron was only found in the case of Fe2TiO5 cathode reduction (Figure 6B) Metallic iron is also seen in 6A -please rephrase

-Page 11: Fe2TiO5 ceramic cathodes were tested in the same experimental conditions but under a galvanostatic mode using 1 A/cm2 current. Change to current density not current.

-Page 11 :  In the present work, while the reddish part of the sample can be related to the initial Fe2TiO5 structure, one can detect the presence of Fe0 in the greyish section (< 1 wt.%), along with Ti-phases (~4 wt.%) attending to Figure 7B.  Is this optical observation confirmed somehow or is it a hypothesis based on the XRD presented? Perhaps reference the expected color of Pseudobrookite

Page 14: use Intermediate phase instead of Indermediary phase

Author Response

The authors would like to thank the reviewer for the valuable comments, which helped us to improve the work. The changes made in the text of the manuscript are marked by track changes.

1. Page 2, when referencing own work (Ref 18) please state it more clearly in the text

The sentence was changed to “The configuration of the bulk working electrode (WE) was similar to the one described as NFAg-R in our previous work [18].”

2. Page 3 "the electric charge passed in the electrochemical cell" is not presented in the paper

The potentials and current density applied in the electrochemical cell were described in the text as requested: “An Autolab PGSTAT302N potentiostat (Metrohm, Switzerland) was used for cyclic voltammetry studies (-1.2 V to 0 V; 10 mV/s) and for the electrochemical reduction under galvanostatic mode (1 A/cm2) or at potentiostatic mode at potentials such as -1.30 V, -1.20 V and -1.15 V.”.

3. Page 4 (and elsewhere) Use of the word "guidelines" is confusing. Consider using 'trend' or similar.

All “guidelines” words were changed for “trends” or “strategies” in the full document.

4. Page 7, section 3.2.2 it is not clear wheather these are the results of a single multi-step experiment or a series of independent experiments. Also the total duration should be mentioned - similar to paragraph 3.2.3.

The following sentences were corrected: “A multi-step experiment was performed by applying a cathodic potential of -1.15 V, followed by an increase to -1.20 V, due to the absence of any noticeable impact on the current density at low polarization (Figure 5A), for a maximum period of 7 h. Relatively low current densities were also reached when a second single experiment was performed, by increasing the cathodic polarization to -1.30 V (< 30 mA/cm2) for 5 h, also showing a noise in the chronoamperometric curve, most likely due to the effects exerted by hydrogen bubbles at the cathode surface.”

Note: the number of Figures was changed due to the addition of one figure in the Materials and Methods section.

5. Page 7: Although exact reasons for such behaviour are still unclear, those might be related to excessive hydrogen evolution in the regions where a good electrical connection with the bulk cathode is still maintained, while another part of the cathode surface become desactivated due to the formation of insulating TiO2 phases. Consider changing those to these and deactivated instead of desactivated.

The words “those” and “deactivated” were changed as requested.

6. Page 7 :Thus, the electrochemical reduction of Fe2TiO5 composition is significantly limited due to the mentioned reasons. Consider changing 'mentioned reasons' to 'described phenomena'.

The term was changed to “described phenomena”.

7. Page 9: when compared with the literature data for porous Fe2O3 samples, please provide reference for literature data.

Literature data was provided in the text as “Current efficiencies were lower, 20% and 6% for      -1.15 and -1.30 V, at 80 °C, when compared with the literature data for porous Fe2O3 samples [8, 15, 18].,”

8. Page 9 : Metallic iron was only found in the case of Fe2TiO5 cathode reduction (Figure 6B) Metallic iron is also seen in 6A -please rephrase.

The authors intended to say that it was only possible to obtain Fe0 at -1.30 V in Fe2TiO5 cathodes and not with lower cathodic polarizations. The Fig. 6A is related to Fe2O3 cathodes, which revealed to show Fe0 regardless the polarization used, as mentioned as “The presence of Fe0 is obvious in the Fe2O3 samples (Figure 7A), where around 19 wt% and 81 wt% of Fe0 are present when the cathodic potentials of -1.15 and -1.30 V (not shown) were applied, respectively.”. For a better understanding, the mentioned sentence was changed to the following: “Concerning the reduction of the Fe2TiO5 cathodes, metallic iron was only observed in this composition when high cathodic polarizations of -1.30 V were applied (Figure 7B).”.

Note: the number of Figures was changed due to the addition of one figure in the Materials and Methods section.

9. Page 11: Fe2TiO5 ceramic cathodes were tested in the same experimental conditions but under a galvanostatic mode using 1 A/cm2 current. Change to current density not current.

The term “current” was changed to “current density”.

10. Page 11: In the present work, while the reddish part of the sample can be related to the initial Fe2TiO5 structure, one can detect the presence of Fe0 in the greyish section (< 1 wt.%), along with Ti-phases (~4 wt.%) attending to Figure 7B.  Is this optical observation confirmed somehow or is it a hypothesis based on the XRD presented? Perhaps reference the expected color of Pseudobrookite.

The mentioned colours are based on the optical observations (Figure 8A), which were later confirmed by the XRD diffractogram shown in Figure 8B. For better clarification, the mentioned sentence was changed in the text to “In the present work, while the reddish part of the sample can be related to the initial Fe2TiO5 structure (red is the initial colour of pseudobrookite samples), one can detect the presence of Fe0 in the greyish section (< 1 wt.%), along with Ti-phases (~4 wt.%), as confirmed by the XRD results presented in Figure 8B.”. The initial colour of pseudobrookite was also mentioned in the previous sentence for a better understanding.

Note: the number of Figures was changed due to the addition of one figure in the Materials and Methods section.

11. Page 14: use Intermediate phase instead of Indermediary phase.

Intermediate phase was used instead of intermediary phase.

Reviewer 3 Report

You find here enclosed some comments and suggestions about your paper:

  • For a better understanding of the process, a diagram of the apparatus and experimental technique used would be appreciated
  • Page 2, last paragraph: the experimental conditions used, and particularly the NaOH concentration of 10M, were determined in a preliminar tests? Or proceeds for other works? If so, it should be indicated in the text and references included.
  • Page 3, last paragraph:  ... amounted 72.7 and 56.0%...
  • I think it would have been useful to know the chemical composition of the materials used, basically to know the presence of impurities in the materials, which have not been detected by X-ray diffraction.
  • Under what conditions are these cathodes considered to be conductive? What species are present to justify the appropriate level of conduction?
  • Page 5, line 5: “The reduction of Fe(III) to Fe(II) species in all...” I understand that this ions are in solid state because if the electrolyte is NaOH, these ions are not in solution. An explaining about this is appreciated.
  • A more extensive comment on the competition between the evolution of hydrogen and the reduction of the different iron species would be interesting.
  • Subscripts in references: 19,22,23,28,29,33,34,36,38,39,42.

Author Response

The authors would like to thank the reviewer for the valuable comments, which helped us to improve the work. The changes made in the text of the manuscript are marked by track changes.

1. For a better understanding of the process, a diagram of the apparatus and experimental technique used would be appreciated

A schematic figure was added in the Materials and Methods section as Figure 1.

2. Page 2, last paragraph: the experimental conditions used, and particularly the NaOH concentration of 10M, were determined in a preliminar tests? Or proceeds for other works? If so, it should be indicated in the text and references included.

The experimental conditions used were based on literature works. The following sentence was added for a better understanding of the experimental conditions used: “The experimental conditions used during the electrochemical reduction tests were selected based on previous results [8, 9, 18].”.

3. Page 3, last paragraph: ... amounted 72.7 and 56.0%...

We thank the reviewer for showing the typo. In fact, the authors intended to say “72%, 70% and 56%” since each efficiency is related to the chemical compositions of Fe2TiO5, Fe2TiO5·Fe2O3 and Fe2O3 compositions, respectively. The mentioned sentence is now corrected accordingly.

4. I think it would have been useful to know the chemical composition of the materials used, basically to know the presence of impurities in the materials, which have not been detected by X-ray diffraction.

We understand the concern of the reviwer. However, pseudobrookite was not obtained as a mineral from nature, but it was processed from pure chemical precursors such as Fe2O3 and TiO2, from abcr GmbH and Alfa Aesar, respectively. It was done precisely to avoid possible impurities issues during the electrochemical reduction and to ensure a better-controlled medium for the reduction. The same is related to the Fe2O3 and Fe2TiO5.Fe2O3 compositions. The powder precursors used have a purity grade higher than 99.8%, as mentioned in the Materials and Methods section.

5. Under what conditions are these cathodes considered to be conductive? What species are present to justify the appropriate level of conduction?

All cathodes used have relatively low electrical conductivity in the experimental conditions used. In general, Ti cations are expected to improve the electrical conductivity of hematite-based compositions, as mentioned in the manuscript: “One should also consider the improvement of the electrical conductivity of the Fe2O3 cathode (10-14 S/cm for Fe2O3 at room temperature [34–36]) when combined with titanium, e.g. in the form of Fe2TiO5 [29,37–39]“. Attending to the Ref. [34], the conductivity at room temperature of pure Fe2O3 varies from 10-14 S/cm to 0.2 S/cm when adding 1 atomic% of Ti.

6. Page 5, line 5: “The reduction of Fe(III) to Fe(II) species in all...” I understand that this ions are in solid state because if the electrolyte is NaOH, these ions are not in solution. An explaining about this is appreciated.

The redox mechanism involves several stages including the reduction in the solid state of Fe(III) to Fe(II) but also the dissolution of Fe(II) species in the strong alkaline media. The dissolution of Fe(II) to aqueous species such as Fe(OH)3- and HFeO2- are predicted by Pourbaix diagrams. The following sentence in the text elaborates the explanation about the mechanism: “The present results confirm the non-direct reduction of hematite-based ceramics to metallic iron, involving a reduction of Fe(III) to Fe(II) aqueous species, where Fe3O4 is usually a well-established intermediate Fe(III)/Fe(II) phase, in accordance with Pourbaix diagrams [40,41]. Moreover, Fe(II) aqueous species such as Fe(OH)3- and mainly HFeO2- from the reductive dissolution of Fe3O4 might be present at the used temperature. The presence of Fe(OH)2 is debatable at temperatures higher than 65 °C [42] but it is still considered in several research works [9,43,44]. One can also consider the dissolution of some Fe(III) species to Fe(OH)4- anions and its reduction to Fe(OH)3- in such strong alkaline media and temperature (~100 °C) [45].”.

7. A more extensive comment on the competition between the evolution of hydrogen and the reduction of the different iron species would be interesting.

Hydrogen evolution takes place typically around -1.1 V in the experimental conditions tested, as it can be observed by the CV curves shown in Figure 4, where the current density increases considerably to higher cathodic polarizations. The cathodic peak present at around -1.1 V in all CV curves coincide with the metallic iron formation, also in fair agreement with the Pourbaix diagram. Thus, one expects the cathodic current will be simultaneously contributed by the reduction to Fe0 formation and hydrogen formation, while the first reduction step to Fe2+ - containing species takes place at lower cathodic polarization without competing HER. Thus, the following sentence was rephrased in the manuscript: “Generally, the reduction of Fe(III) to Fe(II) species in all the compositions tested takes places around ~-1V (C1) (Figs. 3A,C,E), while the reduction to metallic iron occurs in a superimposed region of the voltammogram associated with the hydrogen evolution reaction (HER) due to the water splitting, above ~-1.1 V. HER actively competes for the cathodic current during Fe0 formation, decreasing considerably the Faradaic efficiencies as observed in several works [8, 9, 13, 14, 19, 20] and it also responsible for the collapse of ceramic cathodes as in Ref. [18].”.

8. Subscripts in references: 19,22,23,28,29,33,34,36,38,39,42.

All chemical formulas in the References section were corrected accordingly.

Reviewer 4 Report

I read the article with interest, but its structure is not transparent. In the introduction, there are citations that need to be specified. A short section of materials and methods is followed by an extensive section of conclusions and discussion. The quality of the presented results is high and clearly processed. The conclusion is general and needs to be specified. I believe your article will be interesting to the journal readers after revision.

I have this comments concerning the content of the article: 

  • modify the structure of the article
  • Sentences, which are given in article, has also to be developed in more detail, especially with regard to the cited individual publications for example: [5–9],  "Fe2TiO5 is the most 
    stable iron-titanate phase with a n-type semiconductor behaviour, showing suitable prop￾erties to be used as a catalyst in biomass gasification [22], photocatalyst [23,24], hydrogen 
    generation material [25], gas sensor [26] or ceramic pigments [27]. Pure phase Fe2TiO5 can 
    be obtained manly by solid-state reaction [23,28,29], hydrothermal synthesis [28] and sol-gel [23,24], among others. " etc. 
  • the conclusion is general and needs to be specified

Author Response

The authors would like to thank the reviewer for the valuable comments, which helped us to improve the work. The changes made in the text of the manuscript are marked by track changes.

1. modify the structure of the article.

We are thankful to the reviewer for this comment and partially agree with it. The manuscript was improved in a general way based on the structure of the Materials Journal and suggestions of other reviewers. In fact, our work employs different approaches and conditions to assess if, eventually, pseudobrookite and related materials can be a promising feedstock for the alkaline electrolytic production of iron. The logics behind some approaches might be not straightforward, if the whole picture is not considered. Thus, we would like to explain better our reasoning for structuring the article. In general, two main approaches exist for electrolytic iron production from iron-bearing feedstock: bulk reduction of ceramic cathodes and reduction from ceramic suspensions. While the second one is more suitable for industrial applications, the bulk approach provides more guidelines regarding the electroreduction mechanisms. Therefore, we focused our primary attention on the bulk reduction, as we consider this feedback more relevant for the audience of Materials Journal. Before the electrochemical experiments, a detailed structural and microstructural characterisation is obviously required, as those properties certainly affect the electrochemical mechanisms (e.g., penetration of the electrolyte to the electrode bulk, etc.). In this respect, it makes sense to study both dense and porous samples, as to a certain extent, extreme cases regarding the individual contribution of the bulk cathode conductivity and electrochemically-active surface area. As a next step, the relevant cathodic potential ranges must be assessed by cyclic voltammetry to guide further electroreduction experiments. For the latter, two modes can be implemented, mainly, potentiostatic (chronoamperometry – current density vs. time) and galvanostatic (chronopotentiometry  - cathodic potential vs. time) regimes. Since we found out that no significant Fe0 formation takes place at the iron reduction onset potentials predicted from CV studies, we shifted our attention to the galvanostatic studies involving relatively high cathodic current densities, which, in turn, required the potentials where a competing HER reaction takes place. In fact, such studies are relevant for the process of simultaneous iron and hydrogen production driven by renewable energy sources, an approach that attracts more attention today. Finally, based on the information obtained from the bulk electroreduction, we performed the electroreduction in suspensions, which is more appealing for the industry. Last but not least, we also gave particular attention to the chemical composition aspects, namely, if partial enrichment in iron content can significantly boost the electroreduction of pseudobrookite, and compared the obtained results to those related to pure hematite studied previously. Based on this, we believe that the proposed article structure and corrections made represent an acceptable way to present such a variety of the data described above.

2. Sentences, which are given in article, has also to be developed in more detail, especially with regard to the cited individual publications for example: [5–9],  "Fe2TiO5 is the most stable iron-titanate phase with a n-type semiconductor behaviour, showing suitable properties to be used as a catalyst in biomass gasification [22], photocatalyst [23,24], hydrogen generation material [25], gas sensor [26] or ceramic pigments [27]. Pure phase Fe2TiO5 can be obtained manly by solid-state reaction [23,28,29], hydrothermal synthesis [28] and sol-gel [23,24], among others. " etc. 

The sentences were improved by the following: “Fe2TiO5 is the most stable iron-titanate phase with an n-type semiconductor behaviour, showing suitable properties to be used as a catalyst in biomass gasification [22] due to its compositional flexibility and redox changes and its capacity to absorb in the visible light spectrum allows it to be used as a photocatalyst [23,24]. It is also used as an electrocatalyst  by designing defects-rich heterostructures[25], in gas sensor [26] since it is as a metal oxide semiconductor material or ceramic pigments [27] due to its presence in glassy coatings and glazes after firing at high temperatures. Synthesis of phase-pure Fe2TiO5 can be performed manly by solid-state reaction [23,28,29], hydrothermal synthesis [28] and sol-gel [23,24], among others, while the design of highly porous Fe2TiO5 ceramics seems to be lacking in literature.”

3. the conclusion is general and needs to be specified.

The conclusion was improved as suggested.